# GFLAN: GENERATIVE FUNCTIONAL LAYOUTS

## ABSTRACT

Automated floor plan generation lies at the intersection of combinatorial search, geometric constraint satisfaction, and functional design requirements—a confluence that has historically resisted a unified computational treatment. While recent deep learning approaches have improved the state of the art, they often struggle to capture architectural reasoning: the precedence of topological relationships over geometric instantiation, the propagation of functional constraints through adjacency networks, and the emergence of circulation patterns from local connectivity decisions. To address these fundamental challenges, this paper introduces GFLAN, a generative framework that restructures floor plan synthesis through explicit factorization into topological planning and geometric realization. Given a single exterior boundary and a front-door location, our approach departs from direct pixel-to-pixel or wall-tracing generation in favor of a principled two-stage decomposition. Stage A employs a specialized convolutional architecture with dual encoders—separating invariant spatial context from evolving layout state—to sequentially allocate room centroids within the building envelope via discrete probability maps over feasible placements. Stage B constructs a heterogeneous graph linking room nodes to boundary vertices, then applies a Transformer-augmented graph neural network (GNN) that jointly regresses room boundaries.

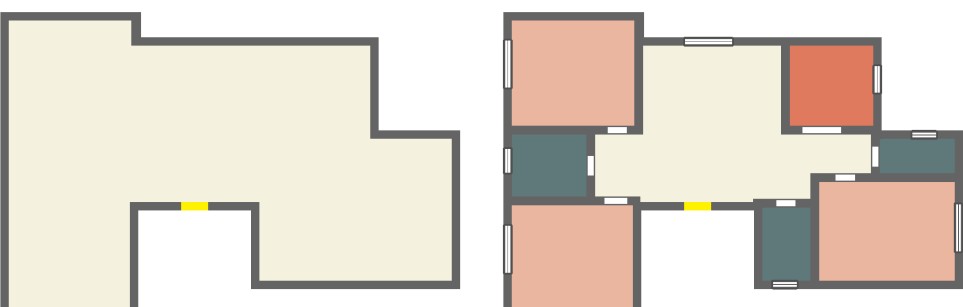

(a) Input: exterior envelope + entrance (yellow).     (b) Output: functional layout from GFLAN.

Figure 1: GFLAN—envelope to layout. Given the exterior envelope and front-door location (a), GFLAN generates a functional floor plan (b) that satisfies adjacency, area, and envelope constraints.

## 1 INTRODUCTION

An architectural floor plan is a scaled, two-dimensional representation of an interior layout that specifies rooms, circulation elements, openings (doors and windows), and their spatial relationships. Beyond a drawing, a floor plan encodes the functional logic of a dwelling: which rooms are adjacent, how occupants move, and how daylight and access are organized. Reliable floor plans are essential for architectural design and for downstream applications such as real-estate visualization, indoor navigation, building simulation, robotics, and gaming environments.

This work addresses data-driven residential floor plan generation. Given a building footprint (the exterior boundary within which rooms must fit) and a high-level functional program—an architectural specification of required spaces (e.g., one kitchen, two bedrooms, one restroom) together with optional adjacency preferences (e.g., "the kitchen near the living room") or access requirements (e.g.,

"each bedroom connects to a hallway")—the task is to produce a valid layout of rooms and openings that satisfies functional and geometric constraints. The desired output is a set of room polygons and openings that lie within the footprint, do not overlap, and realize the requested program.

This problem is challenging for three reasons. *First*, the search space is combinatorial: many distinct layouts can satisfy the same constraints while remaining architecturally valid, and exploring this space while maintaining global consistency is nontrivial. Even coarse abstractions already explode—for $n = 8$ labeled simple graphs, the number of possible room-adjacency graphs is $2^{\binom{n}{2}} \approx 2.68 \times 10^8$ (before any geometry), and assigning room centers on a discrete canvas scales as $O(M^n)$ for $M$ feasible sites (or $M!/(M - n)!$ if each site is unique). Moreover, simplified variants of layout and space allocation reduce to classic NP-hard problems (facility layout; rectangle/space packing), justifying structure-first strategies over exhaustive search (Drira et al., 2007; Pérez-Gosende et al., 2021; Korf, 2003; Chen et al., 2019). *Second*, local choices induce long-range effects: placing a restroom or entry reconfigures circulation and future adjacencies across the plan. *Third*, the layout graph and geometry are tightly coupled (hereafter, "layout graph" denotes the adjacency relations among rooms, to avoid confusion with "topology" in a graph-theoretic sense). Layouts that appear visually realistic can still be invalid, yielding unreachable rooms or sliver spaces—failure modes commonly observed in image-based or wall-tracing pipelines (Sun et al., 2022; Hu et al., 2020).

These issues are addressed with a structure-first, details-second formulation. The method first predicts a compact, discrete description of the plan—room centers inside the given footprint—and only then generates exact room polygons that realize this structure. Making connectivity intent explicit *before* drawing boundaries reduces errors that are costly or impossible to fix later. Concretely, the first stage allocates room centers under program and footprint constraints; the second stage produces axis-aligned room polygons and openings while enforcing footprint containment, non-overlap, and connectivity through global graph-based reasoning over rooms and boundary elements.

This paper contributes three main elements. It formulates floor plan synthesis as a two-stage process that separates connectivity intent (room placement and adjacencies) from geometric realization (room polygons and openings), aligning the learning problem with how designers reason about layouts. It presents GFLAN, a graph-conditioned generative model that infers room centroids and adjacencies from minimal inputs and then generates geometry that satisfies containment and connectivity constraints without hand-crafted post-processing. It evaluates the approach on residential layouts derived from RPLAN (Wu et al., 2019), comparing to representative baselines under each method's native input conditions (e.g., layout graphs for Graph2Plan; fixed-footprint settings for WallPlan) (Hu et al., 2020; Sun et al., 2022). Across a suite of functional metrics, GFLAN reduces connectivity failures and mis-specified adjacencies while maintaining competitive geometric accuracy.

## 2 RELATED WORK

Automatic floor plan generation has been an active research topic in computer graphics and architectural computing for decades (Mitchell, 1975; Eastman, 1971; Grant, 1972; Krawczyk & Dudnik, 1973; Cytryn & Parsons, 1976; Foulds & Robinson, 1976; Blaser & Schauer, 1977; Galle, 1981; Shaviv, 1986). Early approaches relied on procedural rules or optimization techniques to arrange rooms within a boundary. The advent of large-scale datasets such as RPLAN (Wu et al., 2019) and ResPlan (Abouagour et al., 2025) enabled data-driven models to learn common spatial patterns. In recent years, a variety of deep generative models have been proposed specifically for residential floor plan generation (Nauata et al., 2020; 2021; Luo et al.; Upadhyay et al., 2023; Hu et al., 2020; Para et al., 2021; Sun et al., 2022; Dupty et al., 2024; Aalaei et al., 2023; Tang et al., 2023; Shabani et al., 2023; Su et al., 2023; Zeng et al., 2024; Gueze et al., 2023).

**GAN-based models.** Generative adversarial networks (GANs) were among the first deep learning methods applied to this problem. Nauata et al. (2020) introduced HouseGAN, a relational GAN that generates layouts conditioned on a graph of room adjacencies,. Its successor, HouseGAN++ (Nauata et al., 2021), refines this approach and additionally learns to place doors between rooms. Luo et al. proposed FloorplanGAN, which represents floor plans in a vector format (rooms and walls as polygons) while using a raster-based discriminator to improve geometric fidelity. Upadhyay et al. (2023) developed FloorGAN, an end-to-end GAN that directly outputs plausible layouts without post-processing. Other variants incorporate architectural priors: Aalaei et al. (2023) use a graph-constrained conditional GAN to enforce functional adjacencies during generation, and Tang et al. (2023) introduce a Graph Transformer GAN that better captures the adjacency/connection structure

of house layouts. Rahbar et al. (2022) explore a two-stage strategy: first generating a high-level "bubble diagram" of room connections, then refining it into a detailed plan with a conditional GAN. While GAN-based approaches can produce visually plausible layouts, they often rely on predefined adjacency structures or struggle to guarantee all human-centric design criteria without additional constraints. For instance, ensuring every bedroom is near a restroom or maintaining realistic room proportions often requires explicit rules or iterative refinement beyond what basic GANs achieve.

**Graph-based and sequential models.** Beyond image-centric GANs, several methods generate floor plans by treating them as graphs or by sequentially adding architectural elements. Hu et al. (2020) pioneer a graph-based pipeline: given an input layout graph of rooms (nodes) and adjacencies (edges), a GNN coupled with a CNN translates this abstract plan into a spatially explicit floor plan. Similarly, Para et al. (2021) use constraint graphs and a Transformer-based generator to incrementally construct feasible layouts that satisfy adjacency constraints. Focusing on wall-by-wall synthesis, Sun et al. (2022) generate one wall segment at a time by traversing a graph of room boundaries, inherently enforcing boundary continuity, connectivity, and accessibility (i.e., ensuring rooms are reachable via doors/openings). Most recently, Dupty et al. (2024) formulate layout generation as a factor-graph problem, capturing pairwise relations (adjacency, alignment, etc.) between rooms to maintain a coherent configuration. These graph-centric approaches explicitly encode functional spatial relationships—for example, they can enforce that a hallway node connects to multiple rooms or that living and dining areas share a boundary. By generating layouts stepwise (room-by-room or wall-by-wall), they promote valid circulation pathways and adjacencies, though errors can accumulate if local decisions drift from the global design intent.

**Diffusion-based models.** Diffusion models have recently shown strong performance in layout generation. Shabani et al. (2023) introduce HouseDiffusion, a diffusion-based generator operating on a vector floor plan representation; by combining discrete and continuous denoising steps, it can precisely control room locations and sizes (down to wall and door coordinates), yielding layouts with more accurate proportions and alignments. Su et al. (2023) propose a diffusion model for residential design that generates realistic floor plan graphs from scratch, implicitly capturing common room groupings (e.g., clustering bedrooms near restrooms). Zeng et al. (2024) further demonstrate conditional diffusion: their system incorporates multiple design conditions (fixed room counts, area requirements, adjacency constraints) and still produces valid plans, highlighting diffusion's flexibility in satisfying functional criteria. Hybrid approaches also exist; for instance, Gueze et al. (2023) integrate a GNN with constrained diffusion to reconstruct floor layouts from partial observations, promoting graph-consistent connectivity and room accessibility. Overall, diffusion-based approaches excel at modeling the joint distribution of room arrangements and geometry, helping maintain global consistency without an explicit discriminator.

Across all these approaches, ensuring truly functional spaces remains a central challenge. Many authors note that plausibility alone is insufficient—the layout should reflect how people use and move through the space. Important qualitative criteria include placing kitchens near living/dining areas, grouping bedrooms adjacent to restrooms for privacy, keeping room sizes proportional, and providing clear circulation routes via hallways or open spaces. Some models encode these principles via hard constraints (e.g., adjacency requirements in (Nauata et al., 2020; Para et al., 2021; Aalaei et al., 2023)), or on learned priors (e.g., co-occurrence patterns captured by generative models). Nevertheless, even state-of-the-art systems can violate subtle design logic—for example, isolating a bedroom without a hallway or yielding an oversized living room. Recent surveys highlight the need for human-aligned criteria in automated layout synthesis (Weber et al., 2022; Mostafavi et al., 2024). This work introduces GFLAN, a model aimed at advancing floor-plan generation by explicitly incorporating spatial-reasoning principles. By synthesizing layouts that respect adjacency norms and circulation requirements from the outset, GFLAN moves beyond prior GAN- or diffusion-based approaches toward more usable floor plans.

# 3 METHODOLOGY

This section details a two-stage generative approach. Floor plan synthesis is cast as a deterministic two-stage process: (1) sequential room-center placement via convolutional heatmap prediction, and (2) simultaneous room-rectangle regression on a hybrid room–boundary graph. This decomposition separates high-level spatial planning from geometric realization.

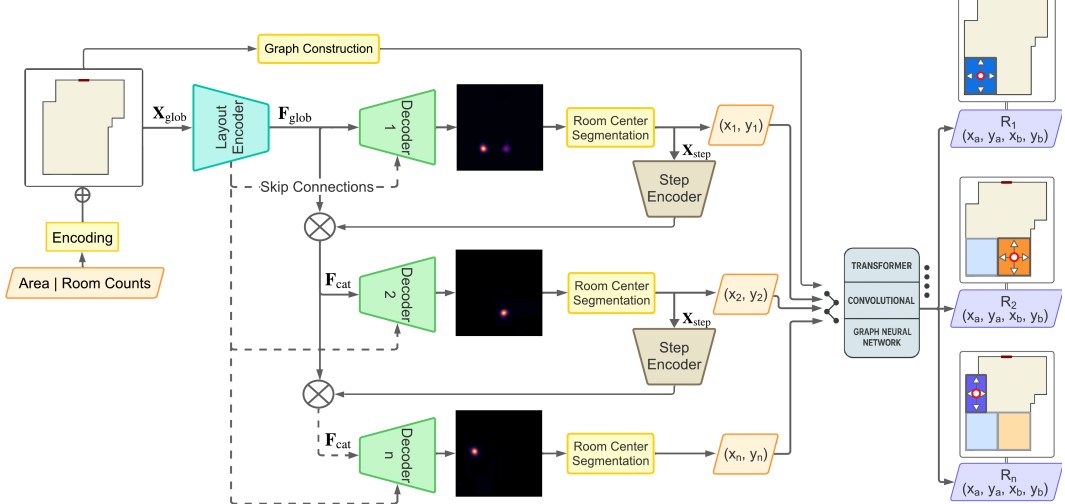

Figure 2: Overview of the two-stage pipeline. A convolutional layout encoder produces latent representations from the building envelope and room program. Sequential decoders then predict room centers conditioned on previously placed rooms. A transformer-based graph neural network regresses full room boundaries while respecting structural constraints.

**Notation.** Let $H = W = 256$. The notation $[H] \times [W] = \{0, \ldots, H-1\} \times \{0, \ldots, W-1\}$ is used for pixel indices and denotes pixel-center coordinates by $\mathbf{x}_{uv} = (u+\frac{1}{2}, v+\frac{1}{2}) \in \mathbb{R}^2$ (image axes). For a point $\mathbf{p} \in \mathbb{R}^2$, $\delta_{\mathbf{p}} \in \{0, 1\}^{H \times W}$ is a binary image with a $1$ at the rasterized pixel of $\mathbf{p}$ and $0$ elsewhere. Set operators $\cap, \cup, \setminus$ act on planar polygons; when needed for raster computations the usual polygon rasterizer is applied to $\{0, \ldots, 255\}^2$. The room-type set is $\mathcal{T} = \{\text{bed}, \text{rest}, \text{kit}, \text{bal}\}$, and $\text{onehot}(t) \in \{0, 1\}^{|\mathcal{T}|}$ denotes a type indicator. The term "px" is used to emphasize pixel units.

### 3.1 PROBLEM FORMULATION

Let $\mathcal{B} \subset \mathbb{R}^2$ be the polygonal building envelope, discretized as a binary mask $\mathbf{B} \in \{0, 1\}^{H \times W}$. The main entrance (front door) is a location $\mathbf{p}_d \in \mathbb{R}^2$ represented by $\delta_{\mathbf{p}_d} \in \{0, 1\}^{H \times W}$. The *architectural program (room counts)* is $\mathbf{c} = (c_1, \ldots, c_T) \in \mathbb{N}^T$ over $T=2$ counted types (bedrooms, restrooms). Unless stated, $\mathbf{c}$ is user-specified.[1] Internally we model four labels (bed, rest, kitchen, balcony); only bed/rest are counted in $\mathbf{c}$, the kitchen count is fixed to one, and balconies are optional exterior attachments.

The generation task is to produce a floor plan $\mathcal{F} = \{\mathcal{R}_1, \ldots, \mathcal{R}_N\}$ of $N = \sum_{t=1}^{T} c_t$ axis-aligned rooms, each parameterized by opposite corners $\mathcal{R}_i = [x_1^{(i)}, x_2^{(i)}] \times [y_1^{(i)}, y_2^{(i)}]$ with $x_1^{(i)} < x_2^{(i)}$, $y_1^{(i)} < y_2^{(i)}$. The remaining envelope area is labeled living, $\mathcal{R}_{\text{living}} = \mathcal{B} \setminus \bigcup_{i=1}^{N} \mathcal{R}_i$.

### 3.2 STAGE A: CNN-BASED CENTER PLACEMENT

#### 3.2.1 SEQUENTIAL GENERATION STRATEGY

A hierarchical placement order based on spatial dominance is used: bedrooms first to set the primary structure, followed by restrooms, then kitchens, and finally balconies (Fig. 2). Each prediction conditions on previously placed rooms via the step-state defined below.

#### 3.2.2 DUAL-ENCODER ARCHITECTURE

**Global Context Encoding.** The global raster $\mathbf{X}^{\text{glob}} \in \mathbb{R}^{H \times W \times C_g}$ encodes plan-invariant inputs

$$\mathbf{X}^{\text{glob}} = [\, \mathbf{B}, \, \delta_{\mathbf{p}_d}, \, \alpha \cdot \mathbf{1}, \, \mathbf{P} \,],$$

---

[1]If $\mathbf{c}$ is absent, an auxiliary predictor infers it from $(\mathbf{B}, \mathbf{p}_d)$; see App. A.

where $\mathbf{1}$ is the all-ones map, $\alpha \in [0, 1]$ is a normalized scalar indicating target floor area (linearly mapped from $50$–$250$ m$^2$), and $\mathbf{P}$ encodes variable program counts. Concretely, bedroom and restroom counts are bucketized and broadcast spatially (four bins per type $\{1, 2, 3, \geq 4\}$, yielding 8 channels). The global encoder $\mathcal{E}_{\mathrm{glob}}$ is DeepLabV3 (ResNet-101):

$$\mathbf{F}_{\mathrm{glob}} = \mathcal{E}_{\mathrm{glob}}(\mathbf{X}^{\mathrm{glob}}) \in \mathbb{R}^{D_g \times h \times w}, \quad h = w = 32, \ D_g = 2048 .$$

**Step-State Encoding.** The step raster $\mathbf{X}^{\mathrm{step}} \in \mathbb{R}^{H \times W \times C_s}$ captures the evolving layout

$$\mathbf{X}^{\mathrm{step}} = \begin{bmatrix} \mathbf{S}_{\mathrm{centers}}, \ \mathbf{S}_{\mathrm{task}} \end{bmatrix} .$$

Here $\mathbf{S}_{\mathrm{centers}} \in \mathbb{R}^{H \times W \times 4}$ has one channel per type; each already-placed center is drawn as a filled disc of radius $r_{\mathrm{ctr}}{=}5$ px. The current-task code $\mathbf{S}_{\mathrm{task}} \in \mathbb{R}^{H \times W \times 4}$ is a spatial broadcast of $\mathrm{onehot}(t_{\mathrm{now}})$. The step encoder $\mathcal{E}_{\mathrm{step}}$ is a second DeepLabV3 (ResNet-101):

$$\mathbf{F}_{\mathrm{step}} = \mathcal{E}_{\mathrm{step}}(\mathbf{X}^{\mathrm{step}}) \in \mathbb{R}^{D_s \times h \times w}, \quad h = w = 32, \ D_s = 2048 .$$

**Feature Fusion.** Concatenation along channels gives $\mathbf{F}_{\mathrm{cat}} = [\mathbf{F}_{\mathrm{glob}}; \mathbf{F}_{\mathrm{step}}] \in \mathbb{R}^{(D_g + D_s) \times h \times w}$.

### 3.2.3 Decoder and Heatmap Generation

**Phase 1: Joint Training.** A shared decoder upsamples $\mathbf{F}_{\mathrm{cat}}$ to a per-type heatmap $\mathbf{H}_t \in [0, 1]^{H \times W}$:

$$\mathbf{z}_1 = \mathrm{ReLU}(\mathrm{BN}(\mathrm{Conv}_{3 \times 3}(\mathbf{F}_{\mathrm{cat}}))) \in \mathbb{R}^{512 \times h \times w},$$
$$\mathbf{z}_2 = \mathrm{ReLU}(\mathrm{BN}(\mathrm{ConvT}_{4 \times 4}^{s=2}(\mathbf{z}_1))) \in \mathbb{R}^{256 \times 2h \times 2w},$$
$$\mathbf{z}_3 = \mathrm{ReLU}(\mathrm{BN}(\mathrm{ConvT}_{4 \times 4}^{s=2}(\mathbf{z}_2))) \in \mathbb{R}^{128 \times 4h \times 4w},$$
$$\mathbf{z}_4 = \mathrm{ReLU}(\mathrm{BN}(\mathrm{ConvT}_{4 \times 4}^{s=2}(\mathbf{z}_3))) \in \mathbb{R}^{64 \times H \times W},$$
$$\mathbf{H}_t = \sigma(\mathrm{Conv}_{3 \times 3}(\mathbf{z}_4)) \in [0, 1]^{H \times W} ,$$

with $\mathrm{ConvT}$ transposed convolution (stride $s$), BN batch norm, and $\sigma$ the logistic sigmoid.

**Phase 2: Type-Specific Specialization.** After Phase 1 early-stops on validation MSE, both encoders ($\mathcal{E}_{\mathrm{glob}}, \mathcal{E}_{\mathrm{step}}$) are frozen, and four identical decoders $\{\mathcal{D}_{\mathrm{bed}}, \mathcal{D}_{\mathrm{rest}}, \mathcal{D}_{\mathrm{kit}}, \mathcal{D}_{\mathrm{bal}}\}$ are fine-tuned independently to specialize spatial priors.

### 3.2.4 Training Objective

Minibatches mix instances uniformly across tasks. Each sample supervises a single ground-truth center $\mathbf{p}^*$ of type $t^*$. The target is a filled disc $\mathbf{T}[u, v] = \mathbb{1}(\|\mathbf{x}_{uv} - \mathbf{p}^*\|_2 \leq r_{\mathrm{ctr}})$, $r_{\mathrm{ctr}} = 5$ px, and the loss is MSE on the corresponding type channel only: $\mathcal{L}_{\mathrm{center}} = \frac{1}{HW} \|\mathbf{H}_{t^*} - \mathbf{T}\|_2^2$, with other type channels ignored for that sample.

### 3.2.5 Peak Extraction

At inference, already-placed centers $\{\mathbf{p}_j\}_{j<k}$ are suppressed by subtracting filled circular masks. Let $\mathbb{1}\{\cdot\}$ be the indicator and $r_{\mathrm{mask}}$ the disk radius (px). Define the binary disk $\mathbf{D}(\mathbf{x}; \mathbf{p}_j, r_{\mathrm{mask}}) = \mathbb{1}\{\|\mathbf{x} - \mathbf{p}_j\|_2 \leq r_{\mathrm{mask}}\}$. The suppressed heatmap is $\tilde{\mathbf{H}}_t(\mathbf{x}) = \mathbf{H}_t(\mathbf{x}) - \sum_{j<k} \mathbf{D}(\mathbf{x}; \mathbf{p}_j, r_{\mathrm{mask}})$ with $r_{\mathrm{mask}} = 5$ px.

Multi-scale Laplacian-of-Gaussian (LoG) blob detection on $\tilde{\mathbf{H}}_t$ yields candidate peaks. Deterministic decoding takes the $\arg \max$; stochastic decoding samples from the spatial categorical distribution proportional to $\tilde{\mathbf{H}}_t^{1/\tau}$ with temperature $\tau = 0.7$.

### 3.2.6 Balcony policy (exterior, boundary-attached)

Stage A predicts a balcony heatmap and selects the boundary-adjacent global-maximum blob (top-1). Stage B snaps this site to the nearest envelope edge and regresses the balcony outward along the edge normal. Balcony area is excluded from interior GFA. See Appendix H, Fig. 13.

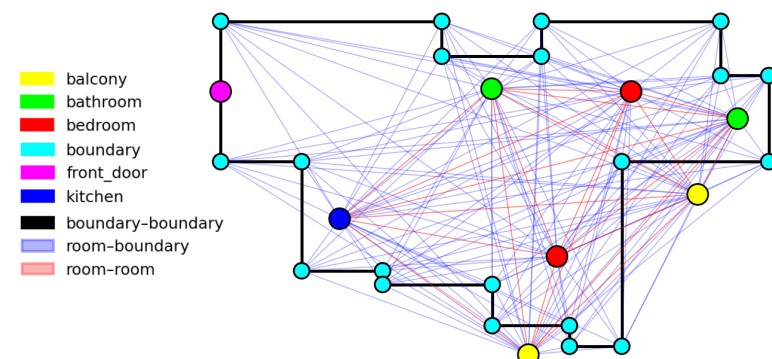

Figure 3: Graph structure used in Stage B. Room nodes connect to neighboring room nodes and nearby boundary nodes. The boundary is discretized into corner/edge nodes.

### 3.3 STAGE B: GRAPH-BASED RECTANGLE REGRESSION

#### 3.3.1 HYBRID GRAPH CONSTRUCTION

From Stage A (3.2) centers $\mathcal{C} = \{(\mathbf{p}_i, t_i)\}_{i=1}^N$, a heterogeneous graph $\mathcal{G} = (\mathcal{V}, \mathcal{E})$ is constructed (Fig. 3).

**Node Set.** $\mathcal{V} = \mathcal{V}_{\text{room}} \cup \mathcal{V}_{\text{bnd}}$, where $\mathcal{V}_{\text{room}}$ contains one node at each $\mathbf{p}_i$, and $\mathcal{V}_{\text{bnd}}$ samples the envelope boundary corners (4–20 nodes depending on outline complexity).

**Edge Set.** For room node $v_i$ with position $\mathbf{p}_i$ (Euclidean norm $\|\cdot\|_2$):

$$\mathcal{E}_i^{\text{bnd}} = \{ (v_i, v_b) : v_b \text{ is one of the 5 nearest boundary nodes to } v_i \},$$
$$\mathcal{E}_i^{\text{room}} = \{ (v_i, v_j) : v_j \text{ is one of the 2 nearest other room nodes to } v_i \}.$$

Ties are broken by index.

**Node Features.** Each node $v$ has an 8D feature $\mathbf{x}_v = \left[ \frac{\mathbf{p}_v}{255}, \ \text{onehot}(t_v) \right] \in \mathbb{R}^8$, where $\mathbf{p}_v \in [0, 255]^2$ and $\text{onehot}(t_v) \in \{0, 1\}^6$ spans the four room types plus two boundary indicators (corner/edge), so $2+6 = 8$.

#### 3.3.2 GRAPH NEURAL NETWORK AND OUTPUTS

A $L=7$-layer TransformerConv GNN updates node embeddings:

$$\mathbf{h}_v^{(\ell+1)} = \text{Dropout}_{0.1}\Big( \text{ReLU}\big( \sum_{u \in \mathcal{N}(v)} \alpha_{vu}^{(\ell)} \mathbf{W}^{(\ell)} \mathbf{h}_u^{(\ell)} \big) \Big),$$

with standard attention weights $\alpha_{vu}^{(\ell)}$ and learned $\mathbf{W}^{(\ell)}$. Hidden widths progress $8 \to 128 \to 256 \to 512 \to 256 \to 128 \to 64 \to 4$. For room nodes, the final 4D output $\hat{\mathbf{r}}_i = (\hat{x}_1, \hat{y}_1, \hat{x}_2, \hat{y}_2) \in [0, 1]^4$ is interpreted as corners after rescaling by 255 and sorting to enforce $\hat{x}_1 < \hat{x}_2, \hat{y}_1 < \hat{y}_2$.

#### 3.3.3 BOUNDARY INTERSECTION (CLIPPING)

Clipping enforces footprint consistency: for interior rooms (bed, rest, kit), $\mathcal{P}_i = \mathcal{R}_i \cap \mathcal{B}$ when $t_i \neq \text{balcony}$; for balconies (exterior, boundary-attached), only the outward part $\mathcal{Q}_j = \mathcal{R}_j \setminus \mathcal{B}$ is retained when $t_j = \text{balcony}$. The assembled plan uses $\{\mathcal{P}_i\}_{t_i \neq \text{balcony}}$ for interior metrics/visualization and overlays $\{\mathcal{Q}_j\}_{t_j = \text{balcony}}$ as exterior attachments. The living area is $\mathcal{R}_{\text{living}} = \mathcal{B} \setminus \bigcup_{t_i \neq \text{balcony}} \mathcal{P}_i$.

### 3.4 TRAINING AND INFERENCE

Both stages are optimized with Adam and cosine annealing. Inference proceeds by sequential center placement with masking, graph construction, rectangle prediction, overlap resolution (corner sorting and non-negative area), and intersection with $\mathcal{B}$. During Phase 2, all encoder parameters remain frozen; only the four type-specific decoders are trainable.

## 4 EXPERIMENTS AND RESULTS

**Setup and baselines.** GFLAN is evaluated on 490 test layouts from RPLAN (Wu et al., 2019), after training on the ResPlan dataset (Abouagour et al., 2025) (17,000 layouts). This protocol tests generalization across datasets. A comparison is made against Graph2Plan and WallPlan, the current state-of-the-art footprint-conditioned methods, with each model evaluated using its intended input format (adjacency graphs for Graph2Plan, footprint only for WallPlan, footprint + program for GFLAN). Reproducibility details (hardware, training settings) are provided in Appendix A. When explicit program counts are unavailable, A lightweight auxiliary predictor (ResNet-18) is employed to estimate bedroom and restroom counts from the footprint mask, normalized area, and front-door location; this module is used only to instantiate inputs and is not part of GFLAN (details in App. A.4).

### 4.1 QUANTITATIVE RESULTS

**Functional Quality Metrics.** Across the suite of functional metrics (Figures 6 and 7), GFLAN shows improved architectural quality. Bedroom-size balance (Fig. 6a) indicates higher min/max bedroom-area ratios for GFLAN (median nearly 0.90) than for WallPlan (around 0.77) or Graph2Plan (around 0.65). Adjacency errors are also lowest for GFLAN: 4 total violations versus 53 for Graph2Plan and 65 for WallPlan (Fig. 7b). The usability ratio is highest for GFLAN (0.82) compared with WallPlan (0.66) and Graph2Plan (0.33) (Fig. 7a). Similarly, the fraction of fully connected layouts is highest for GFLAN (0.95, versus 0.93 for WallPlan and 0.79 for Graph2Plan; Fig. 7c). Full metric definitions are provided in Appendix C.

**Circulation and connectivity.** *Shortest-path sanity* (graph-distance to Euclidean-distance ratio; Fig. 6b) indicates that GFLAN produces more direct routes between rooms: its distribution is shifted lower (closer to 1.0) and tighter than those of both baselines. Together with the higher connectivity ratio (Fig. 7c), this indicates fewer detours and more consistent circulation.

**Program realism.** Figure 5 shows that GFLAN captures realistic room–count distributions, while baselines tend to collapse to narrow modes. In particular, Graph2Plan predominantly outputs single–restroom programs across envelopes (with a small two–restroom tail), whereas GFLAN varies restrooms (1–4) and bedrooms (1–4) in line with the specified program and footprint (see Sec. 3.2.2). This variability better reflects practice, where larger homes typically include multiple bathrooms and bedrooms (metric definitions in App. C).

**Efficiency (runtime & memory).** Average single-sample inference time and memory usage are reported on an NVIDIA A100 GPU (batch=1, $256 \times 256$ raster). As shown in Appendix A.1 (Table 1), GFLAN is $1.71\times$ faster than WallPlan (41.6% time reduction) while using 10.2% less host RAM. Graph2Plan runs faster overall but assumes an adjacency-graph input, whereas WallPlan and GFLAN operate from a raw footprint (plus program for GFLAN). Peak allocated VRAM is also reported; all methods remain lightweight (GFLAN peaks at 1.26 GiB).

### 4.2 QUALITATIVE RESULTS

Figure 4, illustrates characteristic failure modes across four test cases. In scenario (a), Graph2Plan produces an isolated bedroom while WallPlan fragments the living space through poor room placement; GFLAN maintains full connectivity. In scenario (c), Graph2Plan allocates very little area to the kitchen relative to living, while WallPlan creates a windowless kitchen by placing a balcony internally; GFLAN sizes rooms plausibly and ensures the kitchen has exterior access. These patterns are consistent with the quantitative trends above.

## 5 DISCUSSION AND CONCLUSION

The results in Section 4 indicate that factorizing floor-plan synthesis into a topology-first stage (sequential room-centroid placement) and a geometry stage (rectangle regression on a room–boundary graph) better matches architectural reasoning and outperforms baselines under comparable conditions. The topology stage captures adjacency and circulation intent, while the geometry stage instantiates metrically consistent rooms within the fixed envelope (Section 3). This split reduces failure modes typical of monolithic image-to-image or wall-tracing models (e.g., trapped rooms; Fig. 4). Centroid

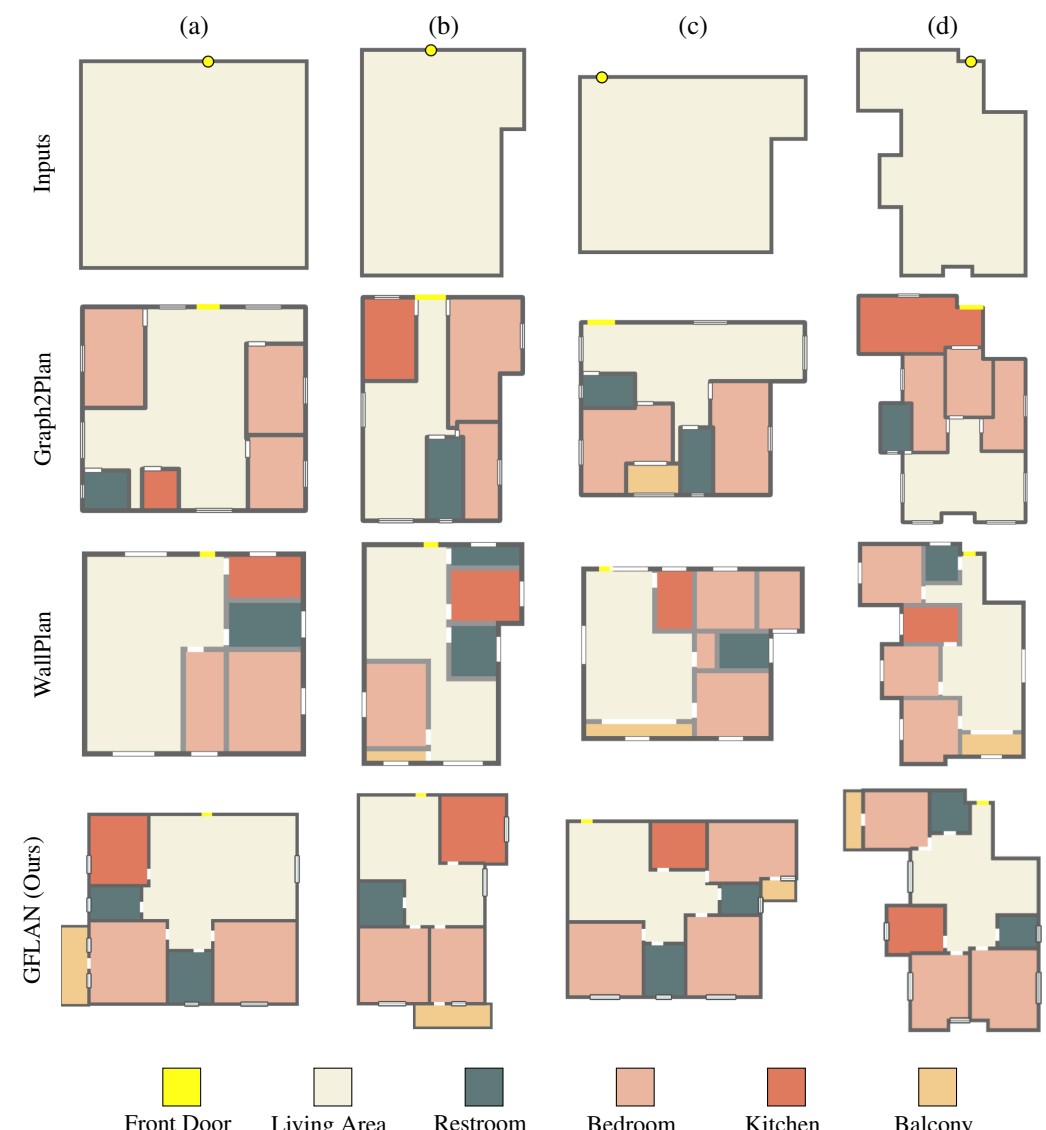

Front Door    Living Area    Restroom    Bedroom    Kitchen    Balcony

Figure 4: Qualitative comparison on four envelopes. Graph2Plan often yields trapped or extreme-aspect rooms and unbalanced zoning; WallPlan can bottleneck circulation or misserved bedrooms (e.g., one restroom for three bedrooms). GFLAN keeps all rooms reachable, preserves near-rectangular shapes, and localizes irregularity to boundary fit. (For the effect of counting balconies as interior floor area, see the "GFLAN-B" ablation in Appendix H, Fig. 13.)

allocation provides an adjacency-aware scaffold that the graph regressor then refines to meet program targets without erasing global intent. Ablation experiments (Appendix B) confirm that both the dual-encoder architecture (global context vs. step-specific encoding) and the two-phase training schedule (freezing encoders before decoder specialization) are critical to achieving the observed improvements in space utilization, circulation efficiency, and program compliance. The framework also remains multi-modal thanks to Stage A's multi-peak heatmaps—sampling different peaks or making small centroid adjustments yields multiple valid layouts for the same inputs. Stage B can realize these variations because it conditions on a hybrid graph rather than absolute pixels. Additionally, GFLAN adapts to variations in the specified entrance location: moving the front door along the envelope yields distinct yet valid layouts that all satisfy the program and connectivity constraints (Appendix E). The use of rectangular room representations prioritizes control and reproducibility; intersecting these rectangles with the envelope yields corner cut-outs or mild L-shapes that adapt to irregular boundaries (Fig. 4).

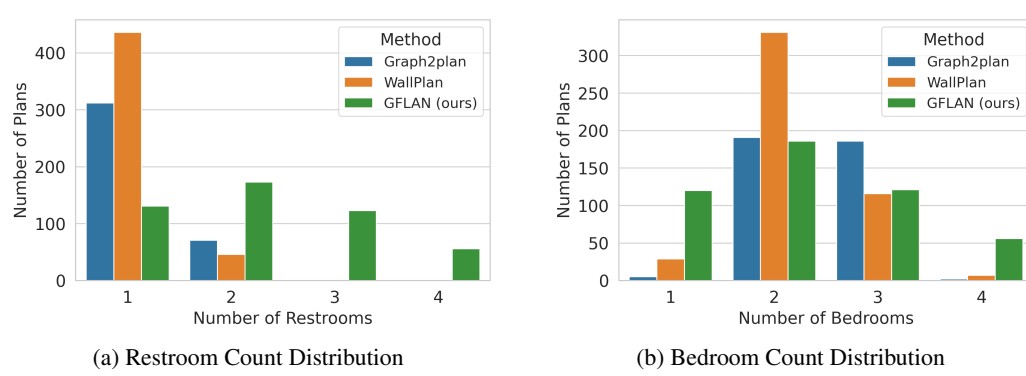

(a) Restroom Count Distribution

(b) Bedroom Count Distribution

Figure 5: Program diversity. GFLAN covers a broader range of restroom and bedroom counts, while Graph2Plan and WallPlan outputs concentrate in a narrow band (e.g., often one restroom).

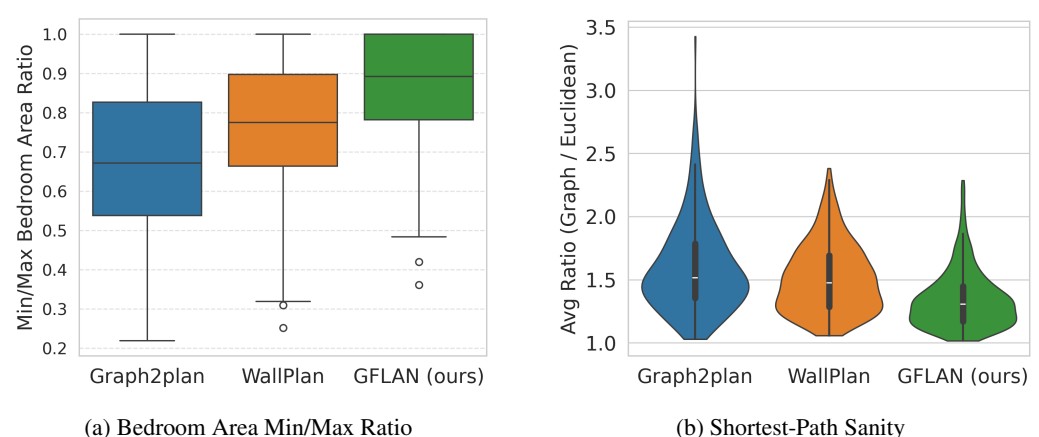

(a) Bedroom Area Min/Max Ratio

(b) Shortest-Path Sanity

Figure 6: Core functional metrics (I). Left: higher min/max bedroom area ratios indicate better bedroom size balance (GFLAN highest). Right: lower graph/Euclidean path ratios indicate more direct circulation (GFLAN lowest and tightest distribution).

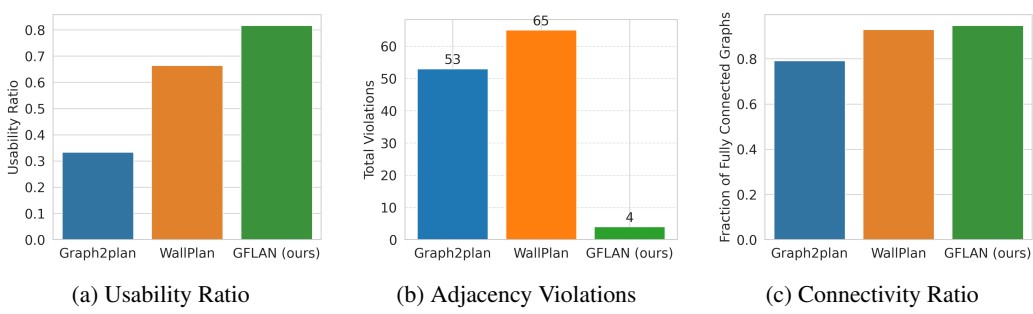

(a) Usability Ratio

(b) Adjacency Violations

(c) Connectivity Ratio

Figure 7: Core functional metrics (II). GFLAN uses space more effectively, satisfies program adjacencies with far fewer violations, and achieves the highest percentage of graph connectivity.

Balconies are modeled as exterior, boundary-attached elements: Stage A locates boundary anchors, and Stage B extrudes each balcony outward; balcony area is excluded from interior floor-area metrics (Appendix H). Limitations: single-story, single-envelope setting; axis-aligned rectangles (no non-orthogonal or curved partitions); residential-only evaluation; no explicit structural/HVAC constraints; potential dataset-bias effects. Future work: multi-story and non-orthogonal geometry, explicit regulatory/constructability constraints, and broader evaluation.

In summary, GFLAN is a lightweight, robust, and efficient generative pipeline that enables the creation of functional designs from minimalistic architectural sketches.

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

# A  IMPLEMENTATION DETAILS

## A.1  INFERENCE EFFICIENCY (RUNTIME & MEMORY)

Runtime and memory are reported for a single-plan (see Table 1); GFLAN achieves competitive efficiency despite its two-stage pipeline. Graph2Plan is faster, but it assumes a supplied room–adjacency graph, and its performance on the quality metrics is lower Fig. 4 and Fig. 7.

Table 1: Inference efficiency (NVIDIA A100; batch=1, $256 \times 256$). Lower is better. GPU column reports peak allocated VRAM.

| Method | Time / plan (s) ↓ | Host RAM (MiB) ↓ | GPU max allocated (MiB) ↓ |
|---|---|---|---|
| Graph2Plan | 0.0799 | 1138.6 | 242.7 |
| WallPlan | 0.2888 | 1477.6 | 709.8 |
| GFLAN | 0.1686 | 1326.9 | 1290.8 |

## A.2  TRAINING CONFIGURATION

**Hardware and Software.**  All experiments were conducted on an NVIDIA A100 GPU using Python 3.10 and PyTorch 1.13. The complete GFLAN training pipeline consumes approximately 13 GiB of GPU memory; inference peaks at ~1.3 GiB (consistent with Table 1).

**Learning Rate Schedule.**  Stage-specific learning rates were employed with cosine annealing schedules for different components. The two-stage training proceeded as follows:

- *Center prediction, Phase 1:* initial learning rate $\eta = 10^{-2}$ for 50 epochs (each epoch $\approx$ 773 steps, batch size 22).
- *Center prediction, Phase 2:* $\eta = 10^{-3}$ for 30 additional epochs (fine-tuning the decoders with encoders frozen).
- *GNN refinement (Stage B):* $\eta = 10^{-4}$ for 100 epochs (training the graph-based regressor with the Stage A encoders fixed).

A cosine annealing schedule with warm restarts ($T_0 = 10$ epochs, $T_{\text{mult}} = 2$) was applied and reset at each phase to help escape local minima during training.

**Data Augmentation.**  To improve generalization, the following augmentations were applied to training samples:

- *Geometric transformations:* random rotation (up to $\pm 15°$), scaling (uniformly in $[0.9, 1.1]$), and translation (up to $\pm 10$ pixels) of the input floor outline.
- *Coordinate jitter:* Gaussian noise with standard deviation $\sigma_{\text{jit}} = 0.03$ (in normalized coordinates) added to each room center during training, making the model robust to slight spatial perturbations.
- Random horizontal and vertical flips with 50% probability each.

These augmentations expanded the base set of 17,000 training floor plans into over 100,000 unique training configurations through random variation.

## A.3  DETAILED GRAPH CONSTRUCTION

The floor plan is represented as a heterogeneous graph $\mathcal{G} = (\mathcal{V}, \mathcal{E})$ combining room nodes and boundary nodes (see Stage B in Section 3.3). This graph is constructed through the following steps:

1. *Polygon extraction:* From the binary floor plan mask, extract the outer boundary polygon with vertices $\{\mathbf{c}_1, \ldots, \mathbf{c}_M\}$ in clockwise order.

2. *Boundary nodes:* Create boundary nodes $\mathcal{V}_{\text{bnd}} = \{v_1^{\text{b}}, \ldots, v_M^{\text{b}}\}$ at these vertices, assigning each $v_i^{\text{b}}$ the coordinates $\mathbf{c}_i$.

3. *Entrance handling:* Let $\mathbf{p}_d$ be the given entrance (front door) point on the envelope. If $\mathbf{p}_d$ coincides with a polygon vertex $\mathbf{c}_j$, label that boundary node $v_j^{\text{b}}$ as the entrance. Otherwise, insert a boundary node $v^{\text{door}}$ at $\mathbf{p}_d$ and connect it to the two polygon vertices between which it lies (say $v_i^{\text{b}}$ and $v_{i+1}^{\text{b}}$).

4. *Perimeter edges:* Connect consecutive boundary nodes to form the outer cycle: $(v_i^{\mathrm{b}}, v_{i+1}^{\mathrm{b}})$ for $i = 1, \ldots, M$ (with $v_{M+1}^{\mathrm{b}} \equiv v_1^{\mathrm{b}}$). No shortcuts or extra perimeter nodes are introduced beyond the original vertices.

5. *Room links:* For each room node $v^{\mathrm{r}}$ (predicted center), add edges linking it to its $k_{\mathrm{bnd}} = 5$ nearest boundary nodes (or all boundary nodes if $M < 5$) and to its $k_{\mathrm{room}} = 2$ nearest other room nodes.

This procedure yields a simple perimeter cycle following the original outline, an entrance node anchored to its incident edge, and sparse room-to-boundary and room-to-room links for efficient message passing. Every room node thus has information about its distance to the exterior and its closest neighbors, enabling the GNN to enforce adjacency and connectivity constraints naturally.

### A.4 AUXILIARY PROGRAM-COUNT PREDICTOR

A compact ResNet-18 is trained to predict bedroom and restroom counts (discrete classes 1–4) from the footprint mask $\mathbf{B}$, normalized area $\alpha$, and front-door location $\delta_{\mathbf{p}_d}$; inputs are stacked as $[\mathbf{B}, \delta_{\mathbf{p}_d}, \alpha \cdot \mathbf{1}]$. The network has two heads (bed, rest), each with cross-entropy loss. It is employed only when counts are not explicitly provided; no gradients flow into GFLAN, and this module does not affect model architecture or metrics.

**Rendering.** Static plots were produced with Matplotlib and Seaborn (Hunter, 2007; Waskom, 2021), while interactive/real-time visualizations and videos were rendered with FURY (Garyfallidis et al., 2021).

**Reproducibility Checklist.** For completeness, key training and evaluation settings are summarized:

- **Environment:** Python 3.10, PyTorch 1.13.1+cu117, CUDA 11.7, cuDNN 8.9, Ubuntu 22.04.

- **Hardware:** NVIDIA A100 40GB for training; NVIDIA RTX 4000 Ada for inference tests.

- **Batch sizes:** Stage A $B_A = 22$ (773 steps/epoch); Stage B $B_B = 32$ (532 steps/epoch).

- **Optimizers:** AdamW ($\beta_1 = 0.9$, $\beta_2 = 0.999$, weight decay $10^{-4}$) for all stages.

- **Learning rates:** Stage A $\eta_0 = 10^{-2}$; Stage B $\eta_0 = 10^{-3}$. Cosine annealing with warm restarts ($T_0 = 10$, $T_{\mathrm{mult}} = 2$) was used, resetting at each phase.

## B ABLATION STUDIES

Additional ablation experiments were conducted to isolate the effects of training strategy and architectural design choices:

**Training Strategy Comparison.** Three different training schedules for the two-stage model were evaluated:

1. *End-to-end from scratch:* Train all components (center placement and GNN) jointly from initialization.

2. *Warm-up then joint:* Train the center-placement module alone for 20 epochs (with the GNN frozen), then unfreeze and continue training everything jointly.

3. *Two-phase (proposed):* The staged strategy described in Section 3, where Phase 1 trains the center-placement stage jointly (all decoders active), Phase 2 then freezes the encoders and fine-tunes separate decoders for each room type, and finally Stage B trains the GNN with encoders still frozen.

Empirically, the two-phase schedule yielded the best results: a higher fraction of valid layouts, fewer room overlaps, lower variance across random seeds, and faster convergence (roughly 80 total epochs vs. 100–120 for the other strategies).

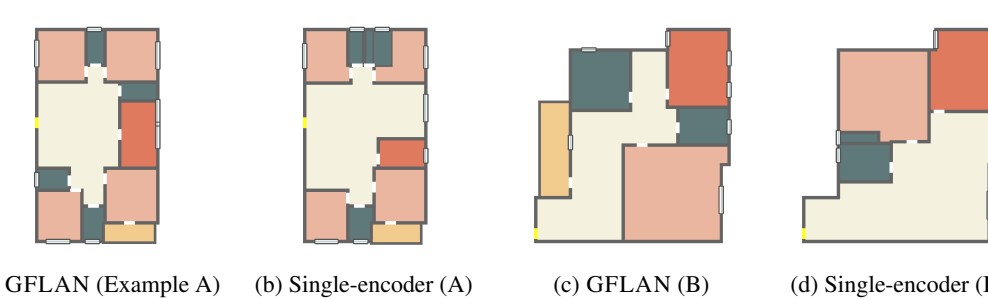

(a) GFLAN (Example A)      (b) Single-encoder (A)      (c) GFLAN (B)      (d) Single-encoder (B)

Figure 8: Ablation of encoder design. Left: dual-encoder model (separate global and step encoders). Right: single shared encoder variant for the same inputs. The single-encoder design produces fragmented circulation, uneven room sizes, and poorly aligned openings, whereas the dual-encoder design yields continuous circulation and cleaner, more space-efficient layouts.

**Graph Connectivity Variants.**   The graph construction hyperparameters in Stage B were also varied to assess their impact (related to the hybrid graph described in Section 3.3):

- *Boundary node density:* Boundary nodes sampled approximately every 3 m, 5 m, or 7 m along the envelope. A spacing of about 5 m (connecting each room node to its five nearest boundary nodes) provided the best balance between faithfully capturing complex outlines and keeping the graph computationally efficient.

- *Room–room connectivity:* Each room node connected to its $k = 1, 2$, or $3$ nearest other room nodes. Setting $k = 2$ (each room links to its two closest neighbors by centroid distance) gave sufficient adjacency information to capture primary spatial relationships without over-constraining the network.

The final GFLAN model uses boundary nodes spaced roughly every 5 m and $k = 2$ room–room connections, which yielded the best validation performance.

**Single vs. Dual Encoder.**   Finally, the network architecture was ablated by removing the dual-encoder design. In this variant, a single shared encoder processes all inputs at each step (instead of separate global and step encoders as in Section 3.2.2). The decoder architecture, loss functions, and training schedule were kept identical to the default model. As expected, this change degraded performance. Qualitatively, the single-encoder model tends to: (i) fragment circulation and create narrow, maze-like corridors; (ii) produce less consistent room sizes and weaker alignment to the envelope; and (iii) misplace doors and windows relative to the final room geometry (see Figure 8, right column). In contrast, the dual-encoder model (Figure 8, left column) maintains continuous circulation, allocates space more evenly, and aligns rooms and openings better with the outer walls.

**Note:** Separating global/static context from step-specific/dynamic inputs stabilizes learning and reduces interference between the center-placement and boundary-regression stages, leading to more coherent final plans.

## C   EVALUATION PROTOCOL AND METRICS

**Usability.**   A layout is deemed usable if it (i) contains all required room types specified by the input program, (ii) provides a residual living area $R_{\text{living}} = \mathcal{B} \setminus \bigcup_i \mathcal{R}_i$ of at least $12$ m$^2$, and (iii) is fully connected in terms of access (every room is reachable from the entrance via openings of width $\geq 0.5$ m without passing through any private space like a bedroom or restroom). The connectivity graph is constructed and paths are validated following established extraction/graph analysis procedures (Moradi et al., 2024; Hu et al., 2020). Any layout violating one or more of these conditions is considered unusable.

**Area Realism.**   A distinction is made between building-code minima and the evaluation thresholds used in this work. Under the 2021 International Residential Code, habitable rooms must be at least $6.5$ m$^2$ ($70$ ft$^2$), with kitchens exempt from minimum area and restrooms subject only to fixture clearance rules irc (2021). For consistent benchmarking and functional adequacy, each generated

bedroom is required to be $\geq 9\,\mathrm{m}^2$, restroom $\geq 4\,\mathrm{m}^2$, kitchen $\geq 7\,\mathrm{m}^2$, and living area $\geq 12\,\mathrm{m}^2$ (within $R_{\mathrm{living}}$). These thresholds are based on practice-oriented guidelines used in prior computational layout studies (Wu et al., 2019; Dorrah & Marzouk, 2021; das Neves Martins et al., 2023; Fei & Sun, 2022; Wang et al., 2021).

**Connectivity Ratio.** For each final layout, a graph is constructed where nodes represent all rooms (plus the entrance) and undirected edges connect any two rooms that share a doorway or sufficiently large opening (at least 0.5 m of open wall). Reachability from the entrance node is then checked. The connectivity ratio is defined as the fraction of rooms reachable from the entrance. A fully connected layout has a connectivity ratio of 1.0. Privacy constraints are enforced as well: a valid circulation path cannot require traversing a bedroom to reach a common area.

**Adjacency Satisfaction.** Many residential programs imply that certain rooms should be directly connected (e.g., each bedroom should open to a common living area). If an input adjacency graph is provided (as in Graph2Plan), an adjacency violation is counted whenever two rooms that should be adjacent (per the program) have no direct opening between them in the generated layout. In the absence of an explicit adjacency list, typical adjacency requirements are approximated by connecting each room to its $k = 4$ nearest other rooms (by centroid distance) and checking if each such pair shares a door. This uniform $k$-NN heuristic is applied to outputs of all methods for evaluation and allows for the consistent tallying of unsatisfied adjacencies.

**Circulation Efficiency.** The indirectness of circulation routes is quantified by comparing path lengths to straight-line distances. For each room $\mathcal{R}_i$, let $d_{\mathrm{path}}(\mathrm{entrance}, \mathcal{R}_i)$ be the length of the shortest path from the entrance to that room along the connectivity graph, and let $d_{\mathrm{Euclidean}}(\mathrm{entrance}, \mathcal{R}_i)$ be the straight-line distance from the entrance to that room's centroid. The circulation efficiency is defined as:

$$\eta_{\mathrm{circ}} \;=\; \frac{1}{N} \sum_{i=1}^{N} \frac{d_{\mathrm{path}}(\mathrm{entrance},\ \mathcal{R}_i)}{d_{\mathrm{Euclidean}}(\mathrm{entrance},\ \mathcal{R}_i)}\ ,$$

where $N$ is the number of rooms. Values closer to 1.0 indicate nearly direct access (high efficiency), whereas larger values indicate more circuitous routes. In this analysis, any layout with $\eta_{\mathrm{circ}} > 1.5$ is flagged as having poor circulation (significantly indirect paths).

**Configuration Diversity.** To assess the diversity of layouts the model can produce (important for generative use cases), multiple samples are generated per input and the following are measured:

- The entropy of the distribution of key layout features (e.g., number of bedrooms and restrooms). Higher entropy indicates a wider variety of configurations.

- The average graph edit distance between pairs of generated layout graphs. A larger average edit distance indicates more structural diversity among outputs.

- The count of unique floor-plan graphs (up to isomorphism) obtained from 100 random generations for a given input footprint and program. In practice, nearly all samples are unique, indicating that stochastic generation yields genuinely different designs.

**Spatial Balance.** The balance of each layout is also quantified in terms of space distribution:

- *Private-to-public area ratio:* $\dfrac{\mathrm{Area(bedrooms + restrooms)}}{\mathrm{Area(living\ area + kitchen)}}$. A ratio near 1.0 is desirable, indicating that private and common zones occupy roughly equal areas.

- *Bedroom size equity:* The ratio $\frac{\max(\mathrm{bedroom\ areas})}{\min(\mathrm{bedroom\ areas})}$ among all bedrooms. A value closer to 1 means all bedrooms are similar in size, while a large value indicates one bedroom is much larger than another (an extreme disparity could signal an unbalanced design, unless intentional like a master suite).

- *Aspect-ratio outliers:* The fraction of rooms whose length-to-width ratio exceeds 3:1. Extremely long or narrow rooms are often impractical; layouts with many such rooms are penalized.

## D  ADDITIONAL EXPERIMENTS

One advantage of the two-stage approach is that it naturally enables multiple valid outputs for the same input constraints by sampling different room placement sequences. After the model predicts a heatmap of potential room centers at each step, it is not necessary to pick only the single highest peak—by selecting a different high-probability blob as the next room center, an alternate valid layout configuration can be explored. Many such alternatives remain architecturally sound, differing only in spatial arrangement.

For example, Figure 9 shows the output of the Stage A centroid-prediction network for a given input, visualized as multiple "blobs" indicating likely room-center locations. Each blob represents a distinct plausible position for the next room. By occasionally choosing a lower-intensity blob instead of always taking the strongest response, the model can generate a different layout in subsequent steps. This process can be repeated to produce a variety of designs for the same footprint and program. All these variations still obey the fundamental constraints (full connectivity, proper adjacencies, etc.) because the model's predictions ensure validity at each step.

Figure 10 demonstrates this diversity. Using the same input outline and room-count requirements, seven distinct layouts were generated by randomly selecting among the top-ranked center proposals at each placement iteration. Despite their differences, each plan is plausible and meets all architectural criteria. This showcases GFLAN's flexibility: it can be used not just to obtain a single solution, but to quickly propose many viable design options for a given project program—a valuable feature for early-stage design exploration (see program distributions in Fig. 5 and connectivity gains in Fig. 7c).

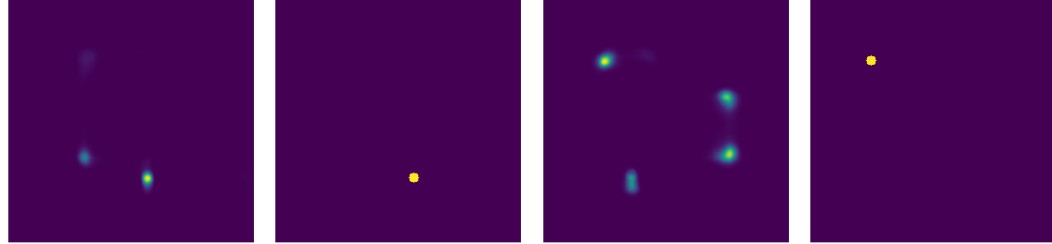

Figure 9: Sample output from Stage A for a given input. The heatmap on the left shows several high-probability "blobs" (bright regions) indicating where a next room could be placed. After applying blob detection, these correspond to distinct potential room centers (right, with one example room type highlighted). Each blob represents an alternative plausible placement for a room. By selecting a different blob (rather than always the single strongest one), a different valid layout is obtained downstream. In practice, the default is to choose the highest-intensity blob for deterministic generation, while stochastic selection enables exploration of multiple design options.

## E  FRONT-DOOR VARIATION STUDY

The model's sensitivity to the placement of the entrance was also evaluated. Because the front-door location is provided as part of the input (Section 3.1), moving it along the envelope while keeping the footprint and program fixed can yield different layouts. Figure 11 shows the result of varying the entrance position for the same building outline and room requirements. Fifteen evenly spaced door locations around the perimeter were tested (at 0%, 5%, 10%, ..., 70% of the outline length). The model produces distinct yet valid floor plans for each different entrance placement, demonstrating that GFLAN adapts to the doorway constraint while preserving overall connectivity and meeting the program requirements.

## F  DOOR AND OPENING INFERENCE FOR CONNECTIVITY

The trained Stage A center generator is also used to infer likely door and window positions via two auxiliary heatmap heads. For each predicted door/window heatmap peak: (i) snap the peak to the nearest wall segment of the predicted room rectangle(s); (ii) place a rectangular opening perpendicular to that wall, with width $\geq 0.5$ m centered at the snapped point; (iii) clip the opening to the wall boundaries; and (iv) assign each door to the two rooms sharing that wall segment (windows attach to

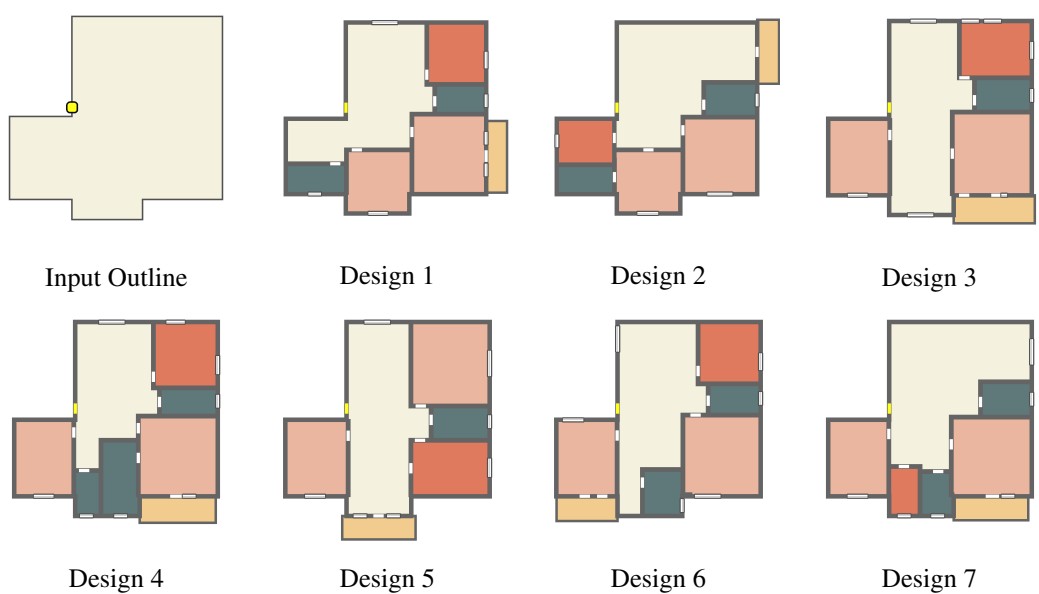

Figure 10: Seven different floor plans generated by GFLAN for the same input footprint and program. Although the room configurations vary across these designs, all layouts are topologically valid and meet the specified program requirements, demonstrating the model's capability to propose diverse yet valid options.

the exterior). Two rooms are marked as connected if and only if a door opening lies along their shared wall. This heuristic yields the connectivity graph used by the metrics in this work; cf. Section 4.1.

## G   ROBUSTNESS TO CENTROID JITTER

**Setup.**   To test robustness, small random perturbations were added to the predicted room centroids before Stage B (polygon regression). Specifically, each centroid $(x, y)$ was translated by $(\delta_x, \delta_y)$ where $\delta_x, \delta_y \sim \mathcal{U}(-\epsilon, \epsilon)$ with noise level $\epsilon \in [0, 5]$ pixels.

**Results.**   Figure 12 shows the effect of centroid jitter. Blue markers indicate the original centroids, and green markers indicate noisy centroids. Despite perturbations, Stage B produced stable, consistent layouts with only minor shifts in room geometry. This demonstrates that GFLAN is insensitive to small centroid jitter, confirming robustness of the graph-based regression.

## H   BALCONIES: EXTERIOR TREATMENT AND BENCHMARKING

**Rationale.**   Balconies are modeled as exterior, envelope-attached elements. They are excluded from interior adjacency, circulation, and gross floor area (GFA). This mirrors common code practice for open balconies and keeps learning focused on interior relationships (q. Huang & h. Zhang, 2005; Puglisi, 2016).

**Modeling and Attachment.**   Let $\mathcal{B}$ be the footprint polygon with boundary edges $\mathcal{E}$. Stage A predicts a balcony heatmap $\mathbf{H}_{bal} \in [0, 1]^{H \times W}$; non-maximum suppression yields boundary-proximal peaks $\mathcal{P}_{bal}$. A single-pass selection is then applied:

$$p^\star = \arg \max_{p \in \mathcal{P}_{bal}} \mathbf{H}_{bal}(p), \qquad e^\star = \arg \min_{e \in \mathcal{E}} \text{dist}(p^\star, e), \qquad a = \Pi_{e^\star}(p^\star),$$

where $\Pi_{e^\star}$ orthogonally projects to edge $e^\star$. Stage B regresses width $w$ and depth $d$ and instantiates

$$\mathcal{R}_{bal} = \text{Rect}\Big(a + \tfrac{d}{2}\hat{\mathbf{n}}_{e^\star}, \, w, \, d, \, \text{axis} \parallel e^\star\Big),$$

then clips to lie strictly outside $\mathcal{B}$. $\mathcal{R}_{bal}$ is omitted from the interior room graph and GFA.

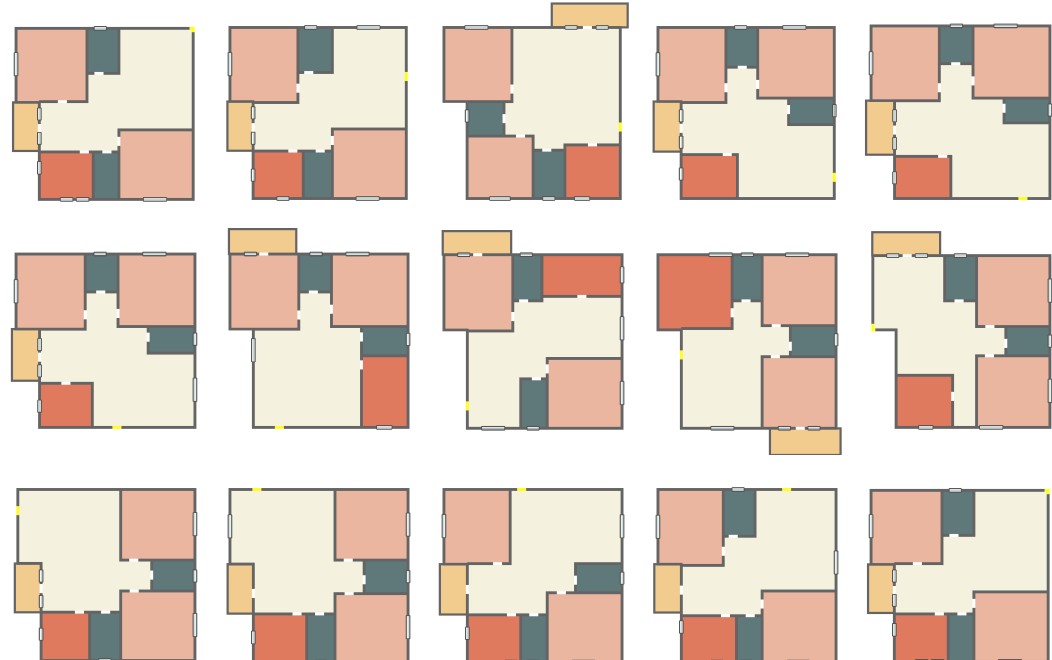

Figure 11: Front-door position sensitivity. Entrance location is varied for the same footprint and program, showing 15 evenly spaced front-door positions. Changing the door location yields distinct yet valid layouts while preserving overall connectivity and satisfying the program constraints.

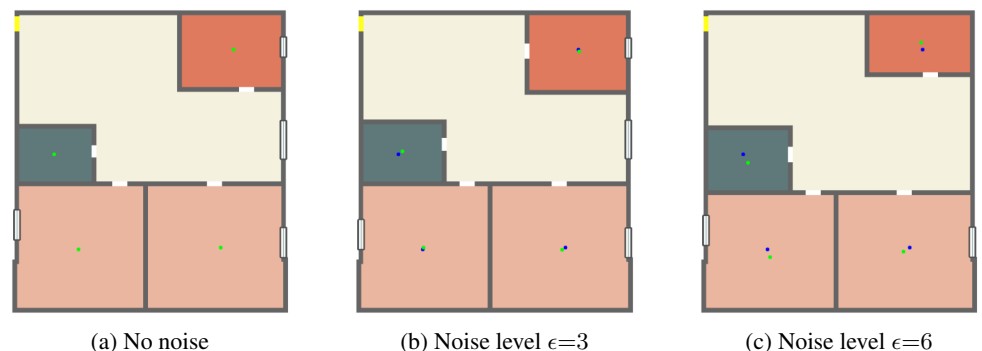

(a) No noise          (b) Noise level $\epsilon=3$          (c) Noise level $\epsilon=6$

Figure 12: **Robustness to centroid jitter.** Original (blue) vs. noisy (green) centroids. Layouts remain stable under centroid perturbations.

**Evaluation Metrics.** Because RPLAN lacks balcony-to-room attachment labels, the evaluation reports: (i) presence/count accuracy; and (ii) boundary-attachment validity, requiring each rectangle to be anchored to some $e \in \mathcal{E}$, not "floating," and with empty intersection with $\mathcal{B}$. Any violation is counted as an error. These checks isolate the exterior-treatment choice from unavailable supervision.

**Ablation (GFLAN-B).** An interior treatment that counts balconies in GFA and adjacency is ablated. This reduces space-utilization scores and yields unrealistic placements (e.g., "balconies" embedded within interior cells). See Fig. 13.

**Scope and Limitations.** The framework supports one or more open balconies attached via the policy above. The model does not infer which interior room a balcony serves, and enclosed sunrooms (which would count toward GFA) are out of scope.

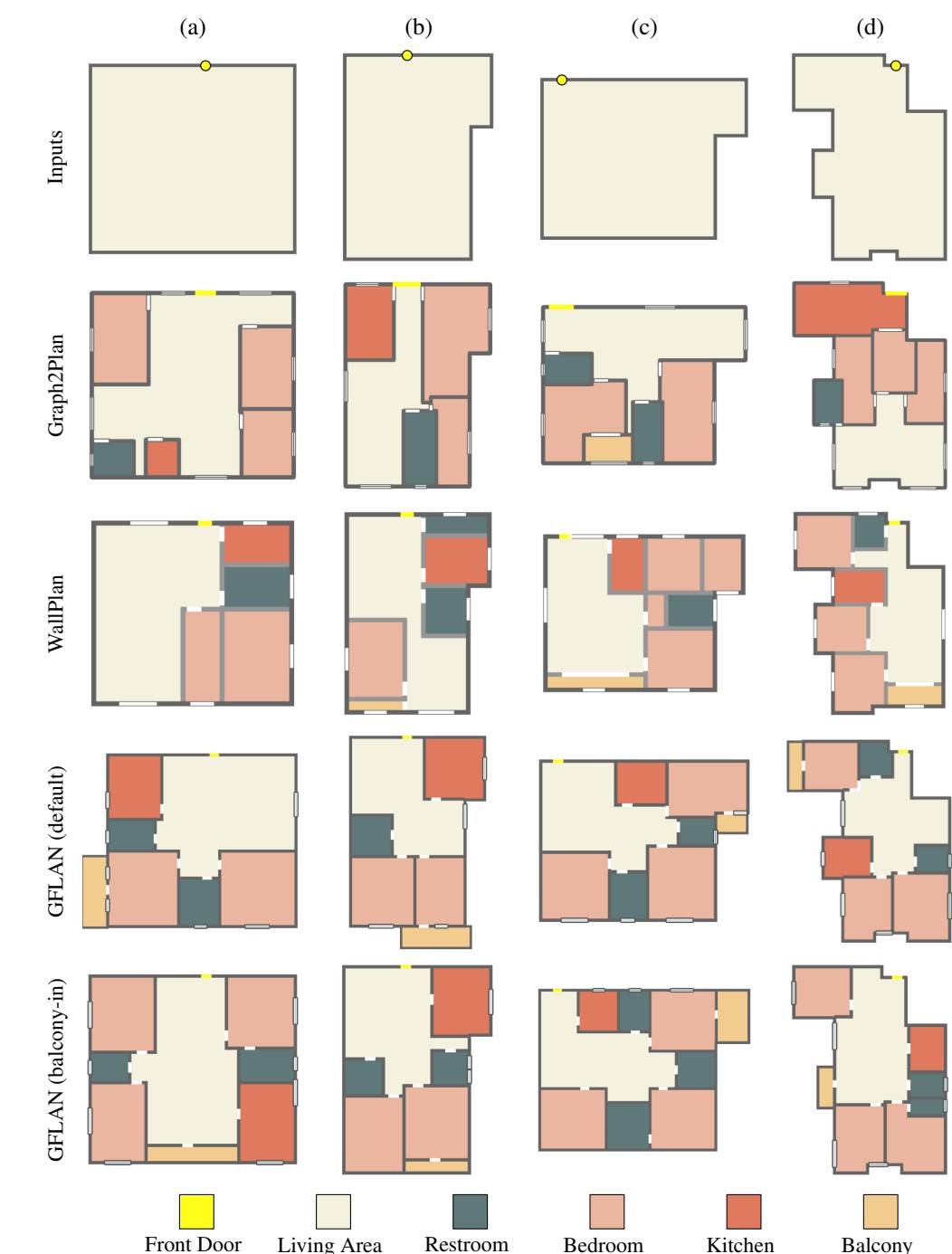

Figure 13: Balcony ablation. Row 5 (GFLAN-B) treats balcony polygons as interior floor area during planning; Row 4 (default) excludes balconies from interior GFA and adjacency.

**Dataset Note.** Because RPLAN does not specify balcony attachments, the single-peak, top-1 policy is applied during benchmarking. A dataset extension with attachment labels would enable supervised, edge-conditioned placement and robust multi-balcony selection.

# I GFLAN FEATURES SUMMARY

Major Contributions:

- **Minimalistic inputs:** The method requires only an envelope polygon, an entrance location, and a room-count program.

- **Two-stage generative pipeline:** A sequential room-center predictor followed by a graph neural network that regresses precise room rectangles, ensuring structural feasibility.

- **Functional floor plans with living spaces:** No trapped rooms. No unrealistic designs.

- **Multiple valid outputs:** The model can propose a variety of layouts by sampling different peaks from the center heatmaps (deterministic mode selects the top peak each time).

- **Fast and lightweight:** Inference is efficient (a fraction of a second per layout on a GPU) and memory-friendly, allowing GFLAN to run on diverse hardware.

- **Robustness to imperfect training data:** The graph stage is trained with jittered room centers, and its bounded refinement allows small adjustments to improve geometry without collapsing the layout.

Additional Contributions:

- **Editable intermediate representation:** Users can optionally add/remove/adjust room centers after Stage A (or even supply an initial set of centers); Stage B then acts as a flexible topology-to-geometry module to complete the floor plan.

- **Balconies as exterior attachments:** Modeled outside the envelope and excluded from interior area calculations; each balcony must attach to an exterior wall segment. GFLAN can adapt to encode balconies as interior too. However, exterior balconies are more common.

GFLAN's free and open-source implementation, experiments and derivative data will be provided upon review completion.

