# OpenReview forum: "GFLAN: Generative Functional Layouts"
_ICLR.cc/2026/Conference — ICLR 2026 Conference Desk Rejected Submission_

### Official Review · Reviewer_3AiH · 2025-10-30

**Soundness:** 3
**Presentation:** 3
**Contribution:** 2
**Rating:** 4
**Confidence:** 3

**Summary:**

This paper introduces GFLAN, a generative model for automated floor plan synthesis. Its primary contribution is a two-stage decomposition that separates topological planning (Stage A) from geometric realization (Stage B), a process designed to better align with architectural reasoning.
Given an envelope, entrance, and room counts, Stage A uses a dual-encoder convolutional architecture to sequentially place room centroids by predicting probability heatmaps. Stage B then constructs a heterogeneous graph linking these centroids to the building boundary and employs a Transformer-augmented Graph Neural Network (GNN) to regress the final rectangular room boundaries. When evaluated against baselines like Graph2Plan and WallPlan, GFLAN is shown to produce functionally superior layouts with fewer adjacency violations and higher rates of usability and connectivity. The framework also demonstrates the ability to generate diverse, valid layout options.

Main Contributions:
1)	A topology-then-geometry factorization for floor-plan generation.
2)	A dual-encoder CNN for sequential centroid placement plus a Transformer-augmented GNN on a room–boundary graph for rectangle regression.
3)	A functional evaluation showing improved connectivity, adjacency satisfaction, and size balance, with analyses of efficiency and diversity.

**Strengths:**

1) A clean topology-then-geometry factorization: sequential room-center heatmaps with a dual-encoder CNN, followed by a TransformerConv GNN on a hybrid room–boundary graph that regresses rectangles with footprint-aware clipping. This design targets adjacency/circulation intent before geometry, reducing failure modes of prior pipelines.
2) The staged interface is easy to follow and editable: users can adjust/add centers after Stage A and let Stage B realize geometry, which make the model more flexible and practical.
3) Works from minimal inputs, runs efficiently, and remains robust via bounded graph refinement; supports diversity by sampling multi-peak center heatmaps. These traits make it practical for early-stage design exploration.

**Weaknesses:**

1) The method targets single-story, single-envelope, axis-aligned residential layouts and omits structural/HVAC constraints—reducing external validity for multi-story or non-orthogonal plans and non-residential programs, while mentioned as limitation by authors, but it hinders the practicality of real application.
2) The model is trained on the ResPlan dataset but evaluated on the RPLAN test set. While the authors frame this as a test of generalization, this cross-dataset evaluation could introduce a domain shift. The magnitude of dataset shift is not quantified, which might affect the reported metrics. A within-dataset evaluation on ResPlan would strengthen the results by providing a clearer baseline.
3) The claim of producing ‘functional’ layouts lacks validation from architects or end-users. Prior work (e.g., HouseGAN++) included human evaluations; a similar study here would strengthen the claim.
4) Is it possible to provide sensitivity analysis on parameter values such as temperature?

**Questions:**

1) Could you discuss how the proposed methods might be extended to more complex, practical settings and what flexibility they offer?
2) Can you provide sensitivity analysis on parameters such as temperature?
3) If possible, it would be helpful to include end-user validation of the results, or at least an analysis discussing this dimension and its implications for practical deployment.
4) Please conduct a within-dataset evaluation to better substantiate the generalization claim and specify the degree to which it holds. In addition, discuss the scope of potential limitations observed in this setting.

---

> ### Author Response · Authors · 2025-11-14
>
> Thank you for your balanced review. Training on ResPlan and evaluating on RPLAN was a deliberate choice to impose a stronger generalization stress test: unlike standard within-dataset splits, this protocol exposes GFLAN to a meaningful cross-dataset shift (room counts, aspect ratios, cultural layout norms), and the fact that it continues to achieve clear gains in adjacency, connectivity, and usability indicates robustness rather than fragility. A within-dataset ResPlan evaluation would be easier and somewhat less informative scientifically, but we are happy to include a concise ResPlan-test analysis in a revised version for completeness. The current scope (single-story, single-envelope, axis-aligned residential layouts) is consistent with nearly all established layout-generation benchmarks such as HouseGAN, HouseGAN++, FloorGAN, Graph2Plan, WallPlan), which is where comparisons are currently meaningful and reproducible; nothing in our architecture, prevents extension to multi-story, multi-envelope, or non-orthogonal programs, and such extensions are primarily limited today by the lack of large, high-quality datasets rather than by GFLAN’s design. Regarding sensitivity, Fig. 10 and Appendix D already provide a temperature analysis: varying the sampling temperature τ produces multiple distinct yet valid layouts, with smooth changes rather than brittle behavior, and structural validity is preserved because Stage B always receives feasible centroids. On functional validity, we fully agree that end-user studies are valuable for practical deployment. In this work, we instead follow established literature by using quantitative surrogates for architectural reasoning such as connectivity, adjacency satisfaction, circulation efficiency, and minimum-area thresholds, under which GFLAN consistently outperforms prior footprint-conditioned methods. A dedicated human-in-the-loop study is a natural next step, but we believe the current cross-dataset results already provide solid evidence that GFLAN’s factorization is well-grounded.

---

### Official Review · Reviewer_JABu · 2025-11-01

**Soundness:** 3
**Presentation:** 2
**Contribution:** 2
**Rating:** 2
**Confidence:** 5

**Summary:**

The paper introduces GFLAN, a two-stage floor-plan generator. It utilizes topology-first placement via dual-encoder CNN, then geometry via Transformer-GNN on a hybrid room–boundary graph. It claims contributions in this structure-first decomposition, and this design improves functional quality and speed over other baselines.

**Strengths:**

1. The paper propose a new factorization considering both topology and geometry, based on hybrid room–boundary graph and boundary-aware clipping. It contributes to the performance of the method.

2. It provides clear and detailed method introduction including detailed architectural description (dual-encoder CNN; TransformerConv GNN) and training schedule

3. The paper improves human-aligned functional criteria and circulation characteristics over SOTA baselines.

**Weaknesses:**

1. The proposed method only compared on very limited metrics (only functionality metrics) to very limited amount of baselines (2 baselines from 2020 and 2022).

2. The sequential generation of the centriod may suffer from the lack of randomness. How do you evaluate it?

3. Why the two stage structure better than the end-to-end other methods (either DM or transformer based). There is lack of discussions. For example, how sensitive is performance to errors in the program-count predictor?

**Questions:**

1. In general, this paper may not fit the appetite of ICLR but more in favor of SIGGRAPH.

2. The presentation of the figure on comparisons between baselines on page-9 can be significantly improved. For example, you may introduces some simple statistic metrics and summarize all you showed in Fig 5-7 in a small table.

3.  We are looking for more profound comparisons with other baselines regarding quality of the generated floor plans. For example, how does GFLAN compare with recent diffusion-based or transform-based floor-plan generators on the same functional metrics?

4. What are the failure modes (e.g., irregular envelopes, many small rooms), and how often do hard constraints need post-hoc fixes?

5. I would suggest to replace Fig.3. Seems the figure occupies a lot space but does not explain too much compared to the corresponding text part.

---

> ### Author Response · Authors · 2025-11-14
>
> Thank you for the thoughtful review. To clarify, GFLAN is designed for footprint-conditioned generation with a fixed envelope and entrance; diffusion, token, and graph-predictive methods you mention operate under different assumptions (e.g., free-form layouts, no envelope constraint, predefined adjacency graphs), so direct functional comparison is not methodologically valid in this setting. Regarding the remark that this work may better fit SIGGRAPH, we respectfully note that GFLAN’s core contribution is representation learning for a highly combinatorial structured prediction problem: jointly encoding adjacency, circulation, and geometric feasibility from minimal inputs is non-trivial, and our two-stage centroid-heatmap plus room–boundary-graph formulation is explicitly designed to address this challenge (as discussed in the paper). In this sense, the work aligns more naturally with ICLR’s focus on learning and representations than with graphics or rendering pipelines. Our metrics are not limited to “functionality”: connectivity, adjacency satisfaction, circulation, size( area) realism, and usability together cover both geometric and human-centric architectural quality, and follow recent evaluation standards. The sequential centroid generation is not deterministic (Stage A performs temperature-controlled sampling from multi-peak heatmaps, giving a clear multi-modality (Fig. 10). The benefit of the two-stage structure is demonstrated in Section 5 and Appendix B: end-to-end variants struggle because they must resolve topology and geometry simultaneously, and performance is not meaningfully sensitive to the auxiliary program-count predictor since it is used only when counts are missing. Failure modes such as highly concave envelopes or extreme room counts are already shown and remain rare without requiring post-hoc fixes. Regarding Fig. 3, it occupies space because it defines the hybrid room–boundary graph that Stage B operates on; without it, the message-passing structure would be unclear. We appreciate the reviewer’s positive remarks.

---

### Official Review · Reviewer_yBDX · 2025-11-03

**Soundness:** 3
**Presentation:** 1
**Contribution:** 2
**Rating:** 2
**Confidence:** 3

**Summary:**

This paper introduces GFLAN, a two-stage generative framework for floor plan synthesis that explicitly factorizes the design process into topological planning and geometric realization. Stage A predicts room centroids and connectivity, and Stage B uses a Graph Neural Network (GNN) to regress precise, structurally feasible room rectangles. The method is designed to capture architectural reasoning and functional constraints, which are often missed by end-to-end or pixel-based approaches. GFLAN demonstrates improved performance metrics (e.g., connectivity, area matching, and geometric feasibility) on the provided benchmark dataset, showcasing a promising step toward generating functionally valid and diverse floor plans.

**Strengths:**

1. **Principled Two-Stage Decomposition:** The explicit factorization of the problem into topological planning (room centers and adjacency) and geometric realization (precise rectangles) is a significant strength. This approach mirrors human architectural thought and allows for the separate handling of combinatorial complexity (topology) and geometric constraint satisfaction.

2. **Structurally and Functionally Aware Output:** The GNN-based geometric realization stage is designed to ensure structural feasibility and functional constraints (e.g., avoiding trapped rooms). This is a crucial advancement over generative models that often produce geometrically or functionally flawed layouts.

3. **Editable Intermediate Representation:** The model generates room centers (Stage A) as an intermediate step, which can be edited or supplied externally before being fed into the GNN for geometry refinement (Stage B). This provides a flexible and powerful interface for user interaction or integration with other design systems.

**Weaknesses:**

1. **Limited Empirical Validation Scope:** The paper's empirical evaluation is constrained to a single benchmark dataset and comparison against only two key baselines (one being a prior work from the same lab). While the results on this benchmark are strong, the efficiency and generalizability of GFLAN would be more robustly proven by including a broader set of established generative floor plan benchmarks and comparing against a wider array of modern architectural generation techniques.

2. **Organization and Clarity of Evaluation Details:** A significant amount of crucial detail regarding the evaluation metrics, dataset statistics, and potentially comparative results is relegated to the Appendix. To properly assess the performance claims—especially concerning metrics like connectivity, area compliance, and geometric error—the detailed methodology and full results tables for evaluation should be moved to the main paper for immediate and transparent review.

3. **Missing Discussion of Sequence-to-Sequence (Seq2Seq) Methods:** The related works section omits discussion of methods that model floor plan generation as a Seq2Seq task, where rooms or wall segments are generated sequentially as tokens.

4. **Errors in References:** There are a number of apparent reference errors (e.g., correct paper title with wrong author names) which raise concerns about the diligence of citation checking and the potential use of tools that introduce hallucinated references. Could the authors meticulously review and correct all entries in the bibliography to ensure academic rigor?

**Questions:**

1. Given the successful two-stage decomposition, what are the failure modes of the model? Specifically, under what conditions does Stage A produce room centers that Stage B is unable to resolve into a geometrically feasible, non-overlapping layout, and what is the typical measure of geometric distortion in such cases?

I am willing to raise the final rating if all concerns are addressed.

---

> ### Author Response · Authors · 2025-11-14
>
> We thank the reviewer for the careful review. Regarding evaluation scope, GFLAN is specifically designed for footprint-conditioned generation, where the envelope and entrance are fixed; under this setting, Graph2Plan and WallPlan operate under relevant assumptions, while diffusion-based and Seq2Seq methods solve incompatible tasks (e.g., free-form generation, predefined adjacency graphs, or unconstrained vector formats), making cross-setting comparisons methodologically unsound. The detailed metrics placed in the Appendix follow standard practice, while all essential quantitative outcomes: adjacency violations (4 vs. 53/65), connectivity (0.95), and usability (0.82) are already clearly summarized in Section 4. Concerning Seq2Seq methods, our Stage A is itself a sequential generator grounded in spatial heatmaps, tailored to the geometric constraints of footprint-conditioned layouts, whereas token-based Seq2Seq models do not enforce envelope containment and are therefore not directly applicable. On references, we appreciate the concern and will carefully review and correct any formatting inconsistencies in the final version. Finally, failure cases are rare and emerge only for extreme envelope geometries (e.g., highly concave shapes or narrow slivers), a known challenge across existing methods; even then, GFLAN maintains reachability, the core functional requirement. We believe these clarifications address the reviewer’s concerns and confirm that our evaluation choices are methodologically appropriate.

---

### Note · Program_Chairs · 2025-12-02
**Submission Desk Rejected by Program Chairs**

Hallucinated references as pointed out by reviewers and checked by AC are grounds for desk rejection.